# Detection and characterization of the SARS-CoV-2 lineage B.1.526 in New York

Anthony P. West Jr. [1✉], Joel O. Wertheim[2], Jade C. Wang [3], Tetyana I. Vasylyeva[2], Jennifer L. Havens[4], Moinuddin A. Chowdhury[3], Edimarlyn Gonzalez[3], Courtney E. Fang[3], Steve S. Di Lonardo[3], Scott Hughes[3], Jennifer L. Rakeman[3], Henry H. Lee[5,6], Christopher O. Barnes[1], Priyanthi N. P. Gnanapragasam[1], Zhi Yang[1], Christian Gaebler[7], Marina Caskey[7], Michel C. Nussenzweig [7,8], Jennifer R. Keeffe [1] & Pamela J. Bjorkman [1]

Wide-scale SARS-CoV-2 genome sequencing is critical to tracking viral evolution during the ongoing pandemic. We develop the software tool, Variant Database (VDB), for quickly examining the changing landscape of spike mutations. Using VDB, we detect an emerging lineage of SARS-CoV-2 in the New York region that shares mutations with previously reported variants. The most common sets of spike mutations in this lineage (now designated as B.1.526) are L5F, T95I, D253G, E484K or S477N, D614G, and A701V. This lineage was first sequenced in late November 2020. Phylodynamic inference confirmed the rapid growth of the B.1.526 lineage. In concert with other variants, like B.1.1.7, the rise of B.1.526 appears to have extended the duration of the second wave of COVID-19 cases in NYC in early 2021. Pseudovirus neutralization experiments demonstrated that B.1.526 spike mutations adversely affect the neutralization titer of convalescent and vaccinee plasma, supporting the public health relevance of this lineage.

[1] Division of Biology and Biological Engineering, California Institute of Technology, Pasadena, CA, USA. [2] Department of Medicine, University of California San Diego, La Jolla, CA, USA. [3] New York City Public Health Laboratory, New York City Department of Health and Mental Hygiene, New York, NY, USA. [4] Bioinformatics and Systems Biology Graduate Program, University of California San Diego, La Jolla, CA, USA. [5] Pandemic Response Laboratory, Long Island City, NY, USA. [6] Department of Genetics, Harvard Medical School, Boston, MA, USA. [7] Laboratory of Molecular Immunology, The Rockefeller University, New York, NY, USA. [8] Howard Hughes Medical Institute, The Rockefeller University, New York, NY, USA. ✉email: apwest@caltech.edu

After the early months of the SARS-CoV-2 pandemic in 2020, the vast majority of sequenced genomes contained the spike mutation D614G (along with three separate nucleotide changes)[1]. Following a period of gradual change, the fourth quarter of 2020 witnessed the emergence of several variants containing multiple mutations, many within the spike gene[2–5]. Multiple lines of evidence support escape from antibody selective pressure as a driving force for the development of these variants[6–9].

Genomic surveillance of SARS-CoV-2 is now focused on monitoring the emergence and spread of these variants and the functional impact that their mutations may have on the effectiveness of passive antibody therapies and the efficacy of vaccines to prevent mild or moderate COVID-19. While an increasing number of specimens are being sequenced, analysis of these genomes remains a challenge[10].

Here, we develop a simple and fast utility that permits rapid inspection of the mutational landscape revealed by genomic surveillance of SARS-CoV-2: Variant Database (vdb). With this tool, we uncover several groups of recently sequenced genomes with mutations at critical antibody epitopes. Among this group is a new lineage emerging in NYC that has increased in frequency to account for ~32% of sequenced genomes in February 2021. We confirm the rapid spread of B.1.526 in NYC during early 2021 through phylodynamic inference. Furthermore, we evaluate the impact of the B.1.526 spike mutations on the neutralization titer of convalescent and vaccinee plasma.

## Results

**vdb**. Phylogenetic analysis is critical to understand the relationships of viral genomes. However, other perspectives can be useful for detecting patterns in large numbers of sequences. We developed vdb as a utility to query the sets of spike mutations observed during genomic surveillance. The vdb program can quickly search SARS-CoV-2 genome datasets with over a million sequences and can be used to explore mutation patterns in various subsets specified by the user. Thus, it can be used to detect emerging mutation patterns that might be overlooked in a conventional phylogenetic analysis examining a fraction of the available sequences. This tool can be used to identify patterns and sequences for further in-depth phylogenetic analysis.

Using the vdb tool to analyze SARS-CoV-2 sequences in the Global Initiative on Sharing Avian Influenza Data (GISAID) dataset[11,12], we detected several clusters of sequences distinct from variants B.1.1.7, B.1.351, B.1.1.248, and B.1.429[2–5] with spike mutations at sites known to be associated with resistance to antibodies against SARS-CoV-2[8,13] (Table 1). The vdb program can find clusters of virus sharing identical sets of spike mutations, and then these patterns can be used to find potentially related sequences.

**Defining mutations of B.1.526**. One notable cluster of genome sequences was collected from the New York region and represents a distinct lineage, now designated as B.1.526 (Fig. 1, Supplementary Fig. 1). This variant is found within the 20.C clade and is distinguished by 3 defining spike mutations: L5F, T95I, and D253G. Within B.1.526, the largest sub-clade is defined by E484K, and two distinct sub-clades are each defined by S477N; both of these mutations located within the receptor-binding domain (RBD) of spike (Fig. 2 and Supplementary Table 1). One of these sub-lineages defined by spike mutations S477N, V701A, and Q957R, has been designated B.1.526.2, but for the purposes of characterizing epidemic and phylodynamic growth of this variant, we combine B.1.526 and B.1.526.2. A closely related lineage, B.1.526.1, is defined by spike mutations D80G, Δ144, F157S, L452R, T859N, and D950H (Supplementary Fig. 2).

We note that the evolutionary history at spike position 701 varies depending on whether the tree is rooted using a molecular clock (as in Fig. 1) versus outgroup rooting with the 20 C ancestor and the clade containing B.1.526.1 (as in Supplementary Fig. 2). The latter rooting posits a substitution A701V followed by a reversion V701A. This uncertainty is not unexpected, as recurrent mutations in spike are common in this part of the phylogeny: both S477N and T95I are homoplastic (Supplementary Fig. 2). Among the nucleotide mutations in lineage B.1.526, the most characteristic include A16500C (NSP13 Q88H), A22320G (spike D253G), and T9867C (NSP4 L438P). Another notable feature of the B.1.526 lineage is the deletion of nucleotides 11288-11296 (NSP6 106-108), which also occurs in variants B.1.1.7, B.1.351, P.1, and B.1.525[14].

Regarding four of the spike mutations prevalent in this lineage: (1) E484K is known to attenuate neutralization of multiple anti-SARS-CoV-2 antibodies, particularly those found in class 2 anti-RBD neutralizing antibodies[13,15], and is also present in variants B.1.351[4] and P.1/B.1.1.248[2], (2) D253G has been reported as an escape mutation from antibodies against the N-terminal domain[16], (3) S477N has been identified in several earlier lineages[17], is near the epitopes of multiple antibodies[18], and has been implicated to increase viral infectivity through enhanced interactions with ACE2[19,20], and (4) A701V sits adjacent to the S2' cleavage site of the neighboring protomer and is shared with variant B.1.351[4]. The overall pattern of mutations in lineage B.1.526 (Fig. 2) suggests that it arose in part in response to selective pressure from antibodies. Based on the dates of collection of these viruses, it appears that the frequency of this lineage has increased rapidly in New York (Table 2).

**Trends in B.1.526 surveillance**. As part of public health surveillance conducted by the New York City Public Health Laboratory (NYC PHL) and the Pandemic Response Lab (PRL) in New York, approximately 13 thousand SARS-CoV-2 genomes

---

**Table 1 Mutation patterns of viruses with mutations at select Spike positions, excluding viruses related to variants B.1.1.7, B.1.351, B.1.1.248, and B.1.429.**

| Pattern | Number of genomes | Top locations | First collection date |
|---|---|---|---|
| L5F T95I D253G E484K D614G A701V | 243 | US(240; NY 235) | 12/16/2020 |
| E484K D614G V1176F | 235 | Brazil(132), US(40) | 4/15/2020 |
| W152L E484K D614G G769V | 49 | US(32) | 11/1/2020 |
| E484K D614G P681H | 37 | US(37; MD 27) | 11/18/2020 |
| R102I F157L V367F E484K Q613H P681R | 36 | England(35) | 12/27/2020 |
| Q52R A67V H69-V70- Y144- E484K D614G Q677H F888L | 36 | England(22) | 12/15/2020 |

Mutations included in this analysis were E484K, N501Y, K417T, K417N, L452R, and A701V. In this table viruses are only included if their spike mutation pattern exactly matches the given pattern. Note about P681H/P681R: variant B.1.1.7 has P681H. Note about W152L: variant B.1.429 has W152C.

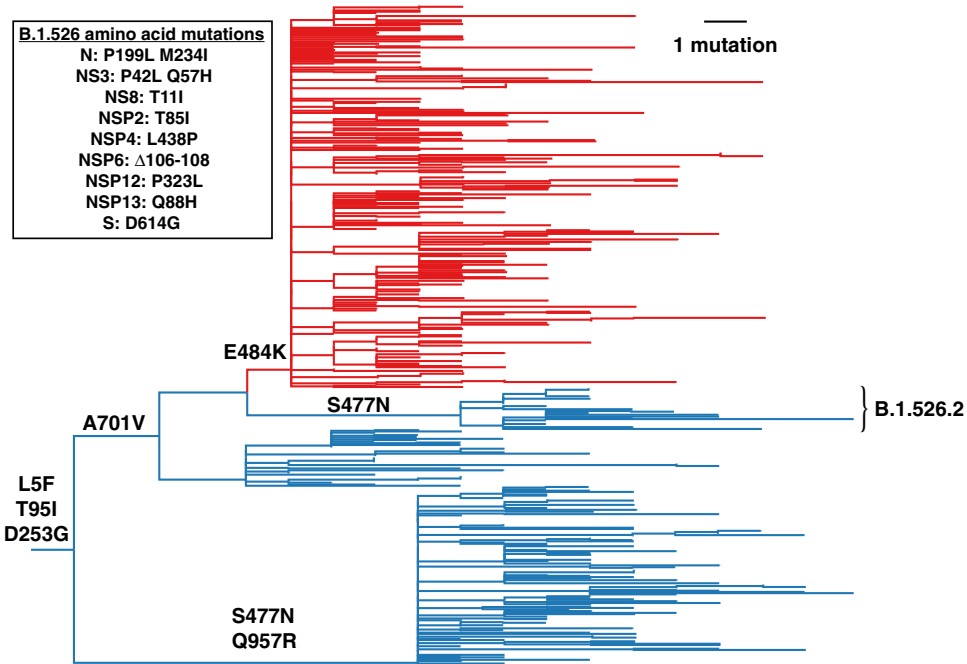

**B.1.526 amino acid mutations**
N: P199L M234I
NS3: P42L Q57H
NS8: T11I
NSP2: T85I
NSP4: L438P
NSP6: Δ106-108
NSP12: P323L
NSP13: Q88H
S: D614G

**Fig. 1 Phylogenetic tree of lineage B.1.526 indicating spike mutations.** Maximum likelihood phylogeny of SARS-CoV-2 variant B.1.526 (including B.1.526.2) sampled by NYC PHL (n = 536). Amino acid substitutions in the spike protein occurring on internal branches are labeled, including the three spike mutations characteristic of B.1.526. The B.1.526 clade defined by the E484K mutation is highlighted in red. Inset highlights non-spike amino acid substitutions and deletions differentiating the B.1.526 clade from the Hu-1 reference genome.

have been sequenced and submitted to GISAID between 1 December 2020 to 29 April 2021. We did not include genomes produced by academic labs in our trend analysis, because viral genomic surveillance by PHL and PRL provide a less biased picture of viral diversity in NYC than other genomes uploaded to GISAID. Of these PHL and PRL genomes, 43.9% are from B.1.526 lineage (including B.1.526.2). The proportion of B.1.526 genomes in NYC has increased since this variant was first detected in NYC surveillance data in late 2020. Its weekly mean exceeded 10% by 14 January 2021 and reached a maximum relative frequency of 57% on 6 March 2021. We used a logistic regression model to capture the trends in relative prevalence of B.1.526, B.1.526.1, and B.1.1.7 (Fig. 3a). The maximum increase in daily prevalence of B.1.526, estimated from the slope at midpoint of the logistic model, took place in January and February 2021. During this time period, B.1.526 increased in prevalence at a rate of 1% per day.

Around 58% (n = 2864) of the B.1.526 genomes contain the E484K mutation, which has also been rising in frequency since early 2021. The weekly mean of B.1.526 genomes with E484K has been above 10% since 01 February 2021, and its maximum increase in daily prevalence inferred with the logistic model was 0.4% per day (Fig. 3a). The increase in frequency of B.1.526 and other variants of interest and concern, such as B.1.1.7, temporally coincides with the peak, subsequent plateau, and slow decline of this epidemic wave in NYC (Fig. 3b, c). From late 2020 to early January 2021, COVID-19 cases and prevalence of B.1.526 (frequency scaled by cases) rose in NYC. However, in early January to late April, the prevalence of B.1.526 continued to rise despite the declining cases over this time (Fig. 3b, c). Only after late April, as cases continued to decline, did the prevalence of B.1.526 fall. This behavior is consistent for both B.1.526 E484 and B.1.526 E484K, as well as B.1.1.7. In contrast, two other variants of interest that had also been detected in NYC in late 2020, B.1.429 and B.1.427, remained at low frequencies throughout this time period (Fig. 3c).

**Geographic distribution of B.1.526 in NYC.** The New York City PHL and the PRL have sequenced 16,901 SARS-CoV-2 genomes from 1 December 2020 through 18 May 2021. Geographic case distribution of specimens received at PHL and PRL for SARS-CoV-2 diagnostic nucleic acid amplification testing (NAAT) are representative of citywide testing efforts. The geographic distribution of over 3300 cases associated with B.1.526 E484K (Fig. 4a) is similar to the distribution of approximately 2400 cases of B.1.526 without the E484K mutation (Fig. 4b). Cases of B.1.526 were widespread across NYC by March with concentrations of cases in northern Manhattan, the Bronx, parts of Queens and Brooklyn. Prior to March 2021, ~90% of B.1.526 genomes deposited to GISAID were from the New York region, but subsequently these lineages have become more prevalent throughout the Northeastern United States.

**Phylodynamic analysis.** During the second epidemic wave in NYC beginning in the fourth quarter of 2020, multiple SARS-CoV-2 variants of concern or interest (i.e., B.1.1.7, B.1.427, and B.1.429) were circulating contemporaneously with the rise of B.1.526 (Fig. 3c). To compare the relative growth rates of these variants during this time-period, we fitted an exponential population growth model[21] implemented in BEAST 1.10.4[22] to the sequences that correspond to these lineages of interest. Specifically, we estimated the exponential growth rate for the B.1.1.7, B.1.427, and B.1.429 variants, for two subsets of the B.1.526 clade sequences (with and without the E484K mutation), and the B.1.526.1 lineage (Fig. 3d). As in the trend analysis, the B.1.526 analysis included the B.1.526.2 subclade.

The B.1.526 E484K clade experienced the most rapid exponential growth compared with other lineages circulating at the same time: 11.82 (95% highest posterior density [HPD]: 9.19–14.54). B.1.526.1, and B.1.1.7 experienced similar growth rates: 9.12 (95% HPD: 5.82–12.39) and 9.29 (95% HPD 7.19–11.41), respectively, followed by B.1.526 with E484: 6.47 (95% HPD: 4.51 – 8.52) The B.1.427 and B.1.429 lineages

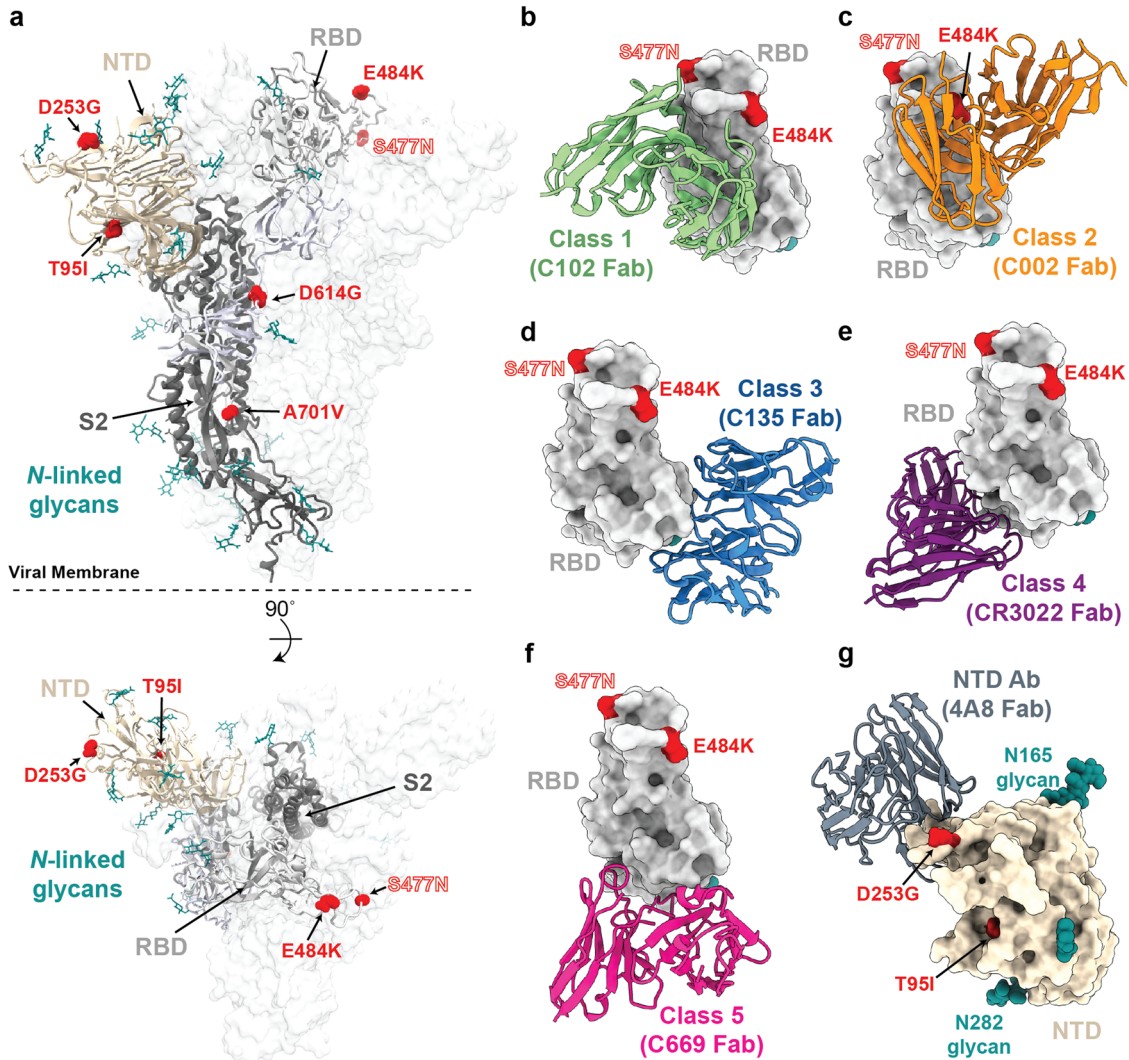

**Fig. 2 Structural locations of the spike mutations of lineage B.1.526. a** Side and top views of the SARS-CoV-2 spike trimer (PDB 7JJI) with mutations of lineage B.1.526 shown as spheres. **b–g** Models of representative neutralizing antibodies (cartoon, VH-VL domain only) bound to RBD (**b–f**, gray surface) or NTD (**g**, wheat surface). Sites for B.1.526 lineage mutations are shown as red spheres, *N*-linked glycans are shown in cyan. The S477N site is also shown (label outlined) for the branch containing this mutation instead of the E484K mutation (see Fig. 1); **b** Class 1 (PDB 7K8M); **c** Class 2 (PDB 7K8S); **d** Class 3 (PDB 7K8Z); **e** Class 4 (PDB 6W41); **f** Class 5[8]; **g** NTD-specific antibody 4A8 (PDB 7C2L).

**Table 2 Counts of virus genomes in lineage B.1.526 by month in New York State.**

| Month | Count | Total sequences | Fraction |
|---|---|---|---|
| Viruses containing spike mutations T95I and D253G (earliest collection date Nov. 23, 2020) | | | |
| Nov. 2020 | 2 | 524 | 0.4% |
| Dec. 2020 | 46 | 2209 | 2.1% |
| Jan. 2021 | 201 | 3148 | 6.4% |
| Feb. 2021 | 1207 | 3868 | 31.2% |
| March 2021* | 124 | 274 | 45.3% |
| Viruses containing spike mutations L5F, T95I, D253G, E484K, D614G, and A701V (earliest collection date Dec. 16, 2020) | | | |
| Nov. 2020 | 0 | | |
| Dec. 2020 | 25 | 2209 | 1.1% |
| Jan. 2021 | 109 | 3148 | 3.5% |
| Feb. 2021 | 628 | 3868 | 16.2% |
| March 2021* | 61 | 274 | 22.3% |

The total number of sequenced genomes examined from GISAID from New York during these time periods is also listed.
*Latest viral collection date was March 4, 2021. Note that geographic sampling may have varied over time as genome sequencing increased.

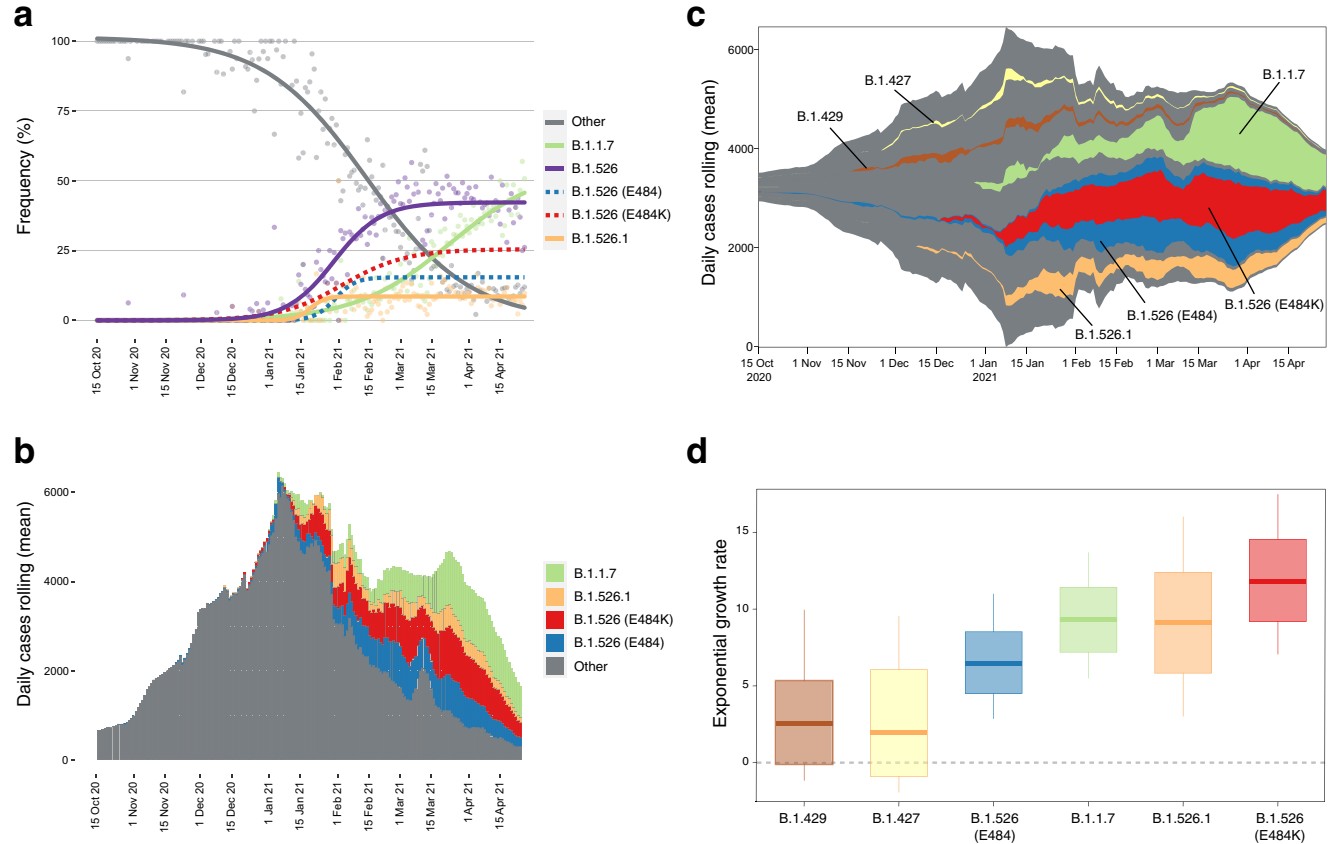

**Fig. 3 Rise of SARS-CoV-2 variants in New York City (NYC) in late 2020 and early 2021. a** Daily frequency (points) and logistic regression trendline for B.1.1.7 (green line), B.1.526 (including B.1.526.2; purple line), B.1.526.1 (orange line), Other (all other sampled SARS-CoV-2 genomes; gray line). Also displayed is the logistic regression for B.1.526 samples with E484K mutation (red dotted line) and without (blue dotted line). **b** Rolling mean number of total daily COVID-19 cases in NYC through time. The color indicates the estimated rolling mean of prevalence out of all samples for B.1.1.7 (green), B.1.526.1 (orange), B.1.526 E484K (red), B.1.526 E484 (blue). **c** Muller plot depicts prevalence of lineages over time displaying rolling mean of daily frequency of lineages scaled by rolling mean of COVID-19 cases. Pseudocounts were added to plot after the emergence of each lineage until last appearance. Nested lineages indicates descendants in phylogeny. **d** Inferred exponential growth rates for SARS-CoV-2 variants in NYC from phylodynamic analysis. The horizontal line indicates the median growth rate estimate, the box denotes the interquartile range, and the whiskers refer to the highest and lowest observations in the posterior sample.

experienced lower growth rates that were not significantly greater than zero: 1.96 (95% HPD: −0.90 to 6.05) and 2.56 (95% HPD: −0.10 to 5.32), respectively. We caution that these lineage growth rates do not distinguish between per-contact transmissibility or per-virion infectiousness and speak only to the relative number of people infected with these variants in NYC during late 2020 and early 2021. Further, a less constrained demographic model (i.e., Bayesian skyline plot) suggests that exponential growth was not maintained across the entire analysis period, as expected following the crest of the second epidemic wave in early 2021 (Supplementary Fig. 3). Nonetheless, the exponential growth rate parameter provides a consistent and interpretable metric by which we can evaluate the growth of SARS-CoV-2 lineages during this time period.

The Bayesian skyline model also allowed us to infer the time of most recent common ancestor (TMRCA) for B.1.526. We estimate the TMRCA of B.1.526 with E484K to be 21 November 2020 (95% HPD: 20 October–13 December), B.1.526 with E484 (including B.1.526.2) to be 7 July 2020 (95% HPD 10 May–30 August), and the B.1.526.1 clades to be 8 November 2020 (95% HPD: 2 October–14 December) (Supplementary Fig. 3).

**Neutralization activity of convalescent and vaccinee plasma against B.1.526.** The identification of several mutations

associated with resistance to anti-SARS-CoV-2 antibodies in B.1.526 sequences raises the question of the impact on SARS-CoV-2 immunity. We generated HIV-based pseudoviruses expressing SARS-CoV-2 spike protein containing either the most common B.1.526 mutation pattern (v.1: L5F, T95I, D253G, E484K, D614G, and A701V), the 2nd most common pattern (v.2: L5F, T95I, D253G, S477N, D614G, and Q957R), or only D614G. Pseudovirus neutralization titers were determined for human plasma samples from vaccinees [Moderna (mRNA-1273) or Pfizer-BioNTech(BNT162b2)][8] or convalescent plasma [at either 1.3[15] or 6.2 months[13] post-infection]. The E484K-containing B.1.526 pseudovirus had a statistically significant reduced neutralization titer compared to the D614G control: for vaccinee plasma, 4.5-fold reduced ($p = 0.00005$); for 1.3-month convalescent plasma, 6.0-fold reduced ($p = 0.03$); and for 6.2-month convalescent plasma, 4.8-fold reduced ($p = 0.02$) (Fig. 5a and Supplementary Tables 2–4). The smaller reduction of the titers in the 6.2-month convalescent plasma samples compared to the 1.3-month samples is consistent with the greater resistance of more matured anti-SARS-CoV-2 antibodies to viral escape mutations[23]. The S477N/Q957R-containing B.1.526 pseudovirus demonstrated a smaller effect on plasma neutralization (Fig. 5b). The 4.5-fold reduced titer of vaccinee plasma against E484K-containing B.1.526 can be compared to an 8.4-fold reduction found against variant B.1.351 (Supplementary Fig. 4 and

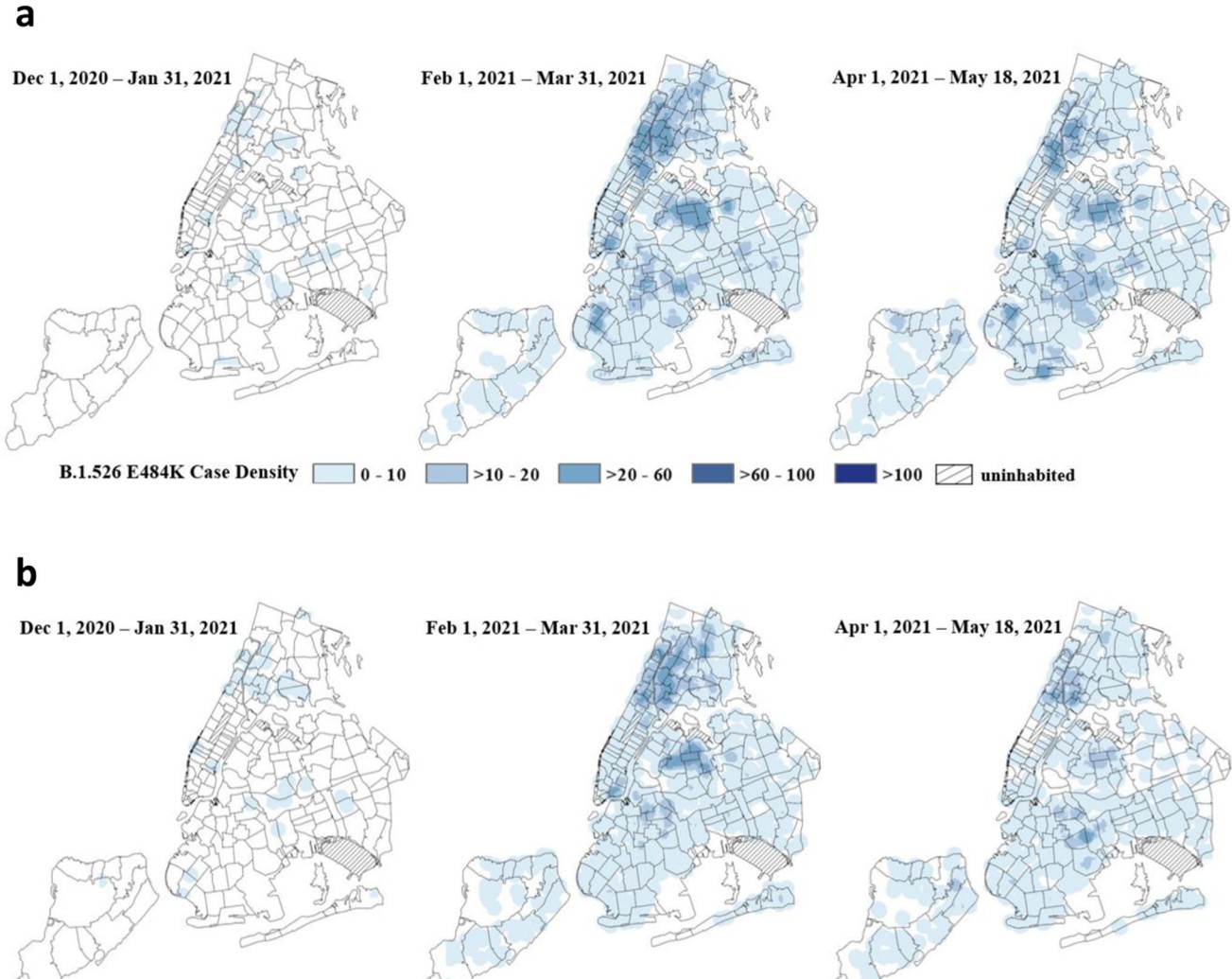

**Fig. 4 Geographic distribution of B.1.526 in New York City. a** Spaciotemporal pattern of B.1.526 E484K lineage in New York City (NYC). Point density of B.1.526 variants with the E484K mutation geo-located by case address overlayed on a map of NYC delineated by zip code. Data for each period is based on specimen collection date. The NYC PHL and the PRL in New York have sequenced 16,901 SARS-CoV-2 genomes from December 1, 2020 thru May 18, 2021 (94.2% of addresses were geocodable). Data represent 38 B.1.526 E484K variants out of 782 sequenced genomes in December 1, 2020–January 31, 2021, 1,755 B.1.526 E484K variants out of 8437 sequenced genomes in February 1–March 31, 2021 and 1,518 B.1.526 E484K variants identified out of a total of 6702 sequenced genomes in April 1–May 18, 2021. **b** Spaciotemporal pattern of B.1.526 without E484K lineage in NYC. Point density of B.1.526 variants without the E484K mutation geo-located by case address overlayed on a map of NYC delineated by zip code. Data for each period is based on specimen collection date. The NYC PHL and the PRL in New York have sequenced 16,901 SARS-CoV-2 genomes from December 1, 2020 thru May 18, 2021 (94.2% of addresses were geocodable). Data represent 45 B.1.526 variants without E484K out of 782 sequenced genomes in December 1, 2020–January 31, 2021, 1,349 B.1.526 variants without E484K out of 8,437 sequenced genomes in February 1–March 31, 2021 and 927 B.1.526 variants without E484K identified out of a total of 6702 sequenced genomes in April 1–May 18, 2021.

Supplementary Table 2). We have evaluated the effect of the B.1.526 (v.1) spike mutations on the neutralization potency of a small panel of anti-SARS-CoV-2 RBD monoclonal antibodies of different classes (Supplementary Table 4). The potency of class 2 antibody C002 was reduced >1000-fold on B.1.526. Antibody C105, a representative of the common VH3-53 public anti-SARS-CoV-2 antibody class, had a 3-fold reduced potency. The other antibodies tested were less affected. These changes are consistent with the effects expected for E484K-containing viruses.

## Discussion

Genomic surveillance is a critical tool to monitor the progression of the COVID-19 pandemic and modeling suggests that

sequencing at least 5% of specimens that test positive for SARS-Cov-2 in a geographic region is necessary to reliably detect the emergence of novel variants at a lower prevalence limit of between 0.1 and 1%[24]. Through the combination of increased sequencing efforts and the use of the software utility described here, we were able to identify the B.1.526 lineage and to begin to characterize its phylogenetic and phylodynamic patterns in NYC in early 2021. Although B.1.526 sequences submitted to GISAID were primarily from New York in January and February 2021, since April 2021 the majority of such sequences were submitted from other US states. The B.1.526 variant has also been described in other recent studies[25,26].

The second epidemic wave in NYC took off in October/November 2020 and peaked in January 2021. However, unlike the

## a

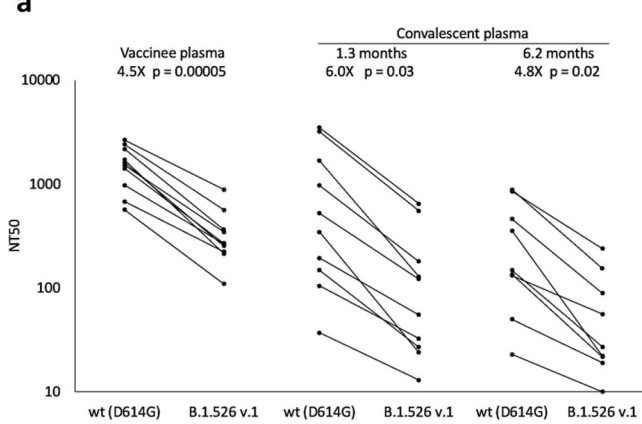

## b

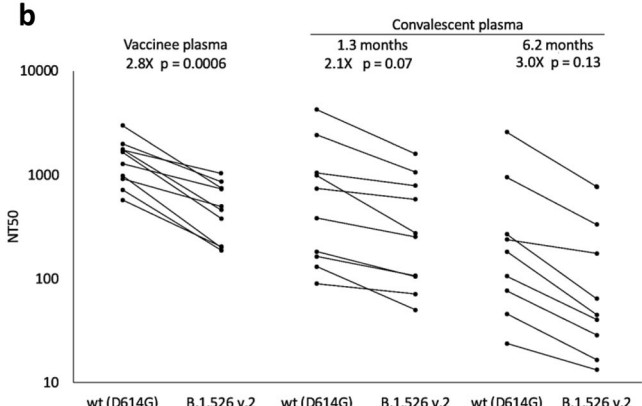

**Fig. 5 Plasma neutralizing activity against pseudoviruses with B.1.526 lineage spike mutations.** SARS-CoV-2 pseudovirus neutralization assays were used to determine neutralization titer (NT50) for COVID-19 vaccinee ($n = 10$) and convalescent plasma at 1.3 months ($n = 10$) and 6.2 months ($n = 9$) after infection. **a** Pseudovirus with spike mutations L5F, T95I, D253G, E484K, D614G (B.1.526 v.1), and A701V. **b** Pseudovirus with spike mutations L5F, T95I, D253G, S477N, D614G, and Q957R (B.1.526 v.2). Statistical significance was determined using paired two-tailed t-tests. Fold differences of means are shown.

first NYC epidemic wave in spring 2020, which had a symmetrical increase and decrease in case counts around its peak, this second epidemic wave was followed by a months-long plateau in the number of reported cases. Remarkably, if we account for the rise of the B.1.526 variant, along with B.1.1.7 and B.1.526.1, the second wave epidemic curve is more symmetric (Fig. 3b). Prior to epidemic peak in January 2021, variants like B.1.526 and B.1.1.7 were at low prevalence; these variants rose to dominance after the second epidemic wave had peaked. Hence, it was the rise of these variants that accounts for the epidemic plateau observed in February/March 2021. These results highlight the importance of SARS-CoV-2 variants in determining the course of the pandemic.

Pseudovirus containing spike gene mutations associated with B.1.526 was significantly more resistant to neutralization by either convalescent or vaccinee plasma. The presence of E484K mutation may play a key role in facilitating increased viral transmission and reducing antibody neutralizing titers, as previously shown in other studies[7,27]. Continued monitoring for emerging variants with mutations such as E484K is important to maximize the impact of public health measures to mitigate the effects of the SARS-CoV-2 pandemic. For example, high frequencies of SARS-CoV-2 variants have potential impacts on selection of appropriate antibody therapeutics and vaccination strategies.

## Methods

**Variant database program**. We developed a software tool named VDB (Variant Database). This tool consists of two Unix command line utilities: (1) vdb, a program for examining spike mutation patterns in a collection of sequenced viral genomes, and (2) vdbCreate, a program for generating a list of viral spike mutations or nucleotide mutations from a multiple sequence alignment for use by vdb. The design goal for the query program vdb is to provide a fast, lightweight, and natural means to examine the landscape of SARS-CoV-2 spike mutations. These programs are written in Swift and are available for MacOS and Linux from the GitHub repository: https://github.com/variant-database/vdb. An online version of vdb is available at http://vdb.live.

The vdb program implements a mutation pattern query language (see Supplementary Method) as a command shell. The first-class objects in this environment are a collection of viruses (a "cluster") and a group of spike mutations (a "pattern"). These objects can be assigned to variables and are the return types of various commands. Generally, clusters can be obtained from searches for patterns, and patterns can be found by examining a given cluster. Clusters can be filtered by geographical location, collection date, mutation count, or the presence or absence of a mutation pattern. The geographic or temporal distribution of clusters can be listed.

Results from vdb presented here are based on a multiple sequence alignment from GISAID[11,12] downloaded on February 10, 2021. Additional sequences downloaded from GISAID on February 22, 2021, were aligned with MAFFT v7.464[28].

**Initial phylogenetic analysis**. Multiple sequence alignments were performed with MAFFT v7.464[28]. The phylogenetic tree was calculated by IQ-TREE v1.6.12[29], and the tree diagram was generated using iTOL v6.1.1 (Interactive Tree of Life)[30]. The Pango lineage nomenclature system[31] provides systematic names for SARS-CoV-2 lineages. The Pango lineage designation for B.1.526 was supported by the phylogenetic tree shown in Supplementary Fig. 1.

**Library preparation and sequencing (NYC PHL)**. For specimens submitted to NYC PHL that tested positive for SARS-CoV-2 with a cycle threshold of below 32 using nucleic acid amplification testing, RNA was extracted from those positive specimens using either the EZ1 (Qiagen, CA), NUCLISENS® easyMAG® (bio-Mérieux Inc., Netherlands), or Kingfisher™ Flex Purification System (Thermo Fisher Scientific, MA) platforms. RNA extracts were subjected to annealing reaction with random hexamers and dNTPs (New England Biolabs Inc., NEB, MA), and reverse transcribed with SuperScript IV Reverse Transcriptase at 42 °C for 50 min. The resulting cDNA was amplified using two separate multiplex PCRs with ARTIC V3 primer pools (Integrated DNA Technologies, IA) per sample in the presence of Q5 2X Hot Start Master Mix (NEB) at 98 °C for 30 s, followed by 35 cycles of 98 °C for 15 s and 65 °C for 5 min[32,33]. The resulting PCR products per sample were combined and purified using Agencourt Ampure XP magnetic beads (Beckman Coulter, IN), at a ratio of 1:1 sample to bead ratio and quantified using a Qubit 3.0 fluorometer (Thermo Fisher Scientific, MA). The PCR products were normalized to 90 ng as input for the NEBNext Ultra II Library Preparation Kit according to standard protocol (NEB): Briefly, the ARTIC PCR products were subjected to simultaneous end-repair, 5'-phosphorylation, and dA-tailing reaction at 20 °C for 30 min, followed by heat inactivation at 65 °C for 30 min. NEBNext Adaptor was then ligated at 25° for 30 min, and then cleaved by USER Enzyme at 37 °C for 15 min. This product was subjected to bead cleanup at a ratio of 0.6x sample to bead ratio. The eluted product was amplified for 6 cycles using NEBNext Ultra II Q5 Master Mix in the presence of NEBNext Multiplex Oligos for Illumina (NEB). The PCR product was purified with Ampure XP beads at a 0.6x sample to bead ratio. The product was a barcoded library containing Illumina P5 and P7 adapters for sequencing on Illumina instruments. The individual libraries were quantified, normalized, and pooled at equimolar concentration and loaded onto the Illumina MiSeq sequencing instrument using V3 600-cycle reagent kits and a V3 flow cell for 250-cycle paired-end sequencing (Illumina, CA).

**Genome assembly**. All raw paired-end sequence reads are trimmed using Trim Galore version 0.6.4_dev[34] removing NEB adapters and quality score below 20 from ends of the reads. The trimmed reads were assembled using the Burrows-Wheeler Aligner MEM algorithm (BWA-MEM) version 0.7.12[35] with SARS-CoV-2 Wuhan-Hu-1 (GenBank accession number MN908947.3) as the reference sequence. Intrahost variant analysis of replicates (iVar)[36] tool was used to remove primer sequences from the amplicon-based sequencing data. Finally, the mutation calls and consensus genome were built using a combination of samtools mpileup[37] and iVar consensus, with a minimum quality score of 20, frequency threshold of 0.6, and minimum depth of 15 to optimize high quality variant calls. A sequence mapping quality control tool developed in-house was used to assess depth of coverage across all sequences, percent of ambiguous bases in the consensus genome and percent sequence mapped to the reference genome. Consensus genome with more than 3% ambiguous bases or less than 95% reference mapped were excluded from any further analyses.

**Library preparation and sequencing (PRL)**. Positive RNA specimens between cycle threshold of 15–30 were selected from all samples tested at Pandemic Response Labs, NYC and cDNA for each specimen was generated using LunaScript RT SuperMix (NEB, MA) according to manufacturer protocol. To target SARS-CoV-2 specifically, cDNA for each specimen was amplified in two separate pools, 28- and 30-plex respectively, to generate 1200 bp of overlapping amplicons[38] using Q5 2x Hot-Start Master Mix (NEB, MA). The resulting pools are combined in equal volume and enriched for full length 1200 bp product using a SPRI-based magnetic bead cleanup. Enriched amplicons are tagmented (Illumina, CA) and barcoded (IDT, IA) and paired-end sequenced on an Illumina MiSeq or NextSeq 550.

**Genome assembly (PRL)**. For each specimen, sequencing adapters are first trimmed using Trim Galore v0.6.6[34], then aligned to the SARS-CoV-2 Wuhan-Hu-1 reference genome (NCBI Nucleotide NC_045512.2) using BWA MEM 0.7.17-r1188[35]. Reads that are unmapped or those that have secondary alignments are discarded from the alignment. Consensus and mutations were called using samtools[37] and Intrahost variant analysis of replicates (iVar)[36] with a minimum quality score of 20, frequency threshold of 0.6 and a minimum read depth of 10× coverage. A consensus genome with ≥90% breath-of-coverage with ≤3000 ambiguous bases is considered a successful reconstruction (as per APHL recommendation).

**Genome alignment**. Complete genome sequences produced by the NYC PHL and the PRL with reported collection dates on or before 29 April 2021 were analyzed. We restricted our analysis to genomes produced by public health surveillance to NYC to reduce bias due to geography or preferential sequencing of viral variants by academic institutions. Genomes were aligned to the Wuhan-Hu-1 reference genome (GenBank Accession MN908947) using mafft v7.475 (mafft --6merpair --keeplength --addfragments)[28]. Pango lineage designations[31] for variants were assigned using Pangolin v2.4.2[39]. Note that Pango designations v.1.2.13 merged the B.1.526 sublineages B.1.526.1 and B.1.526.2 back into the parent lineage.

**Trends in B.1.526 surveillance**. To compare the relative frequency of lineages over time, we calculated daily frequency of each lineage, using PHL NYC and PRL samples from 15 October 2020 to 28 April 2020, 19 days had no samples. The B.1.526 lineage includes B.1.526.2; B.1.526 was separated into a clade defined by E484K mutation (B.1.526 E484K), and samples without this mutation (B.1.526 E484) for some subsequent analysis. We fit daily frequency of each lineage with a logistic model (nls function in R).

To compare the prevalence of lineages over time within the epidemic, we scaled the rolling 7 day mean of daily lineage frequency, by the rolling 7 day mean of confirmed and probable cases. Case count data was obtained from NYC Department of Health and Mental Hygiene (https://github.com/nychealth/coronavirus-data) on 20 May 2021. Prevalence of lineages, and all samples of not specified lineages (Other), were visualized with bar plot (ggplot2) and Muller plot (epiMuller).

**Geocoding addresses**. To identify areas with the highest density of B.1.526 sequenced genomes in NYC from December 1, 2020 to May 18, 2021, patient addresses were geocoded and visualized on a map[40]. Geocoding was performed using the NYC DOHMH's Geoportal application. Once geocoded, a map representing the point locations of individuals with sequenced B.1.526 genomes was created in ArcMap (v. 10.6.1) and exported as a point feature class.

**Point density method**. Point density maps of individuals with B.1.526 sequenced genomes were created by using the point density tool in ArcMap. Point density calculates the density-per-unit area from point features (individuals with a SARS-CoV-2 B.1.526 sequenced genome) that fall within a defined neighborhood by totaling the number of points that fall within the neighborhood divided by the neighborhood area. Density calculations result in the observed gradient patterns. The point density map parameters were 3000 ft radius from the center of 150 square foot cells. The symbology classes for point density classification was set at case densities 0–10, >10–20, >20–60, >60–100.

**Maximum likelihood phylogenetic inference**. Maximum likelihood trees were inferred in IQTree2 using full genome sequences under a GTR+F+$\Gamma_4$ substitution model separately for all genomes sampled by NYC PHL[41]. Minimum branch length of 1e−9 was enforced and an expanded NNI search (--allnni) was employed to improve topology search. This tree was used to select PHL genomes for variant specific phylodynamic inference (see below). To characterize the history of spike mutations in B.1.526, we also inferred two B.1.526 trees under a GTR+F+I substitution model. The first tree comprised B.1.526 and B.1.526.2; the second tree comprised B.1.526, B.1.526.1, B.1.526.2, closely related genomes, and two genomes representing the 20.C ancestor. TreeTime v.0.8.1 was used to root and perform ancestral state reconstruction under a fixed substitution rate of $8 \times 10^{-4}$ substitutions/site/year and a skyline coalescent model[42](Fig. 1). We did not include genomes sequenced by the NYC PRL in the phylogenetic and phylodynamic analysis, because the size of these alignments made standard phylodynamic inference intractable (and the PHL data were sufficiently informative for phylodynamic inference).

**Bayesian phylodynamic inference**. We performed population growth rate inference in coalescence-based framework using an exponential growth model and a Bayesian Skyline model as implemented in BEAST 1.10.4[22]. We used a strict molecular clock model with the fixed substitution rate of $8 \times 10^{-4}$ substitutions/site/year and applied a GTR+F+$\Gamma_4$ substitution model. We specified the OneOnX distribution prior for the population size parameter and Laplace distribution prior (mean = 0.0, scale = 1.0) for the growth rate prior for the exponential growth model. For the Bayesian Skyline analyses, we split the time into 10 intervals and applied uniform population size prior for these time-intervals. Markov chain Monte Carlo analyses were run for 50-300 million generations; the first 10% of samples were discarded as burn-in. Separate inference was performed for each variant: B.1.1.7 ($n = 257$), B.1.427 ($n = 20$), B.1.429 ($n = 40$), B.1.526 E484 (including B.1.526.2; $n = 228$), B.1.526 E484K ($n = 310$), and B.1.526.1 ($n = 143$). For the B.1.526 phylodynamic inference, we did not include two sequences most closely related to B.1.526 (hCoV-19/USA/NY-NYCPHL-001701/2020 and hCoV-19/USA/NY-NYCPHL-002542/2021).

**Human plasma samples**. Human plasma samples were among those collected in previously reported studies[8,13,15]. The study visits and blood draws were performed in compliance with all relevant ethical regulations, written informed consent was obtained, and the protocol for human participants was approved by the Institutional Review Board (IRB) of the Rockefeller University (protocol #DRO-1006).

**Pseudovirus neutralization by human plasma samples and antibodies**. Human plasma samples were assayed for neutralization activity against lentiviruses pseudotyped with SARS-CoV-2 spike containing a 21-amino acid cytoplasmic tail deletion and either D614G or mutations corresponding to lineage B.1.526 (v.1: L5F, T95I, D253G, E484K, D614G, and A701V; v.2: L5F, T95I, D253G, S477N, D614G, and Q957R; constructed using primers listed in Supplementary Table 5). Pseudotyped lentiviruses were generated and neutralizations assays were conducted as previously described[43,44]. Briefly, lentiviral particles were produced in HEK293T cells (ATCC CRL-11268) by co-transfecting the gene encoding SARS-CoV-2 spike protein (D614G or B.1.526) and Env-deficient HIV backbone expressing Luciferase-IRES-ZsGreen. Plasma samples were heat inactivated at 56 °C for 1 h, then 3-fold serial diluted and incubated with SARS-CoV-2 pseudotyped virus for 1 h at 37 °C. The virus/plasma mixture (or plasma/antibody mixture) was added to 293T$_{ACE2}$ target cells (J. Bloom laboratory, Fred Hutchinson Cancer Research Center), which were seeded the previous day on poly-L-lysine coated plates. For antibody assays the top concentrations were 10 µg/mL (BG10-19 and BG7-15) or 50 µg/mL (C105 and C102), with 3-fold serial dilutions. After incubating for 48 h at 37 °C, target cells were lysed with Britelite Plus (Perkin Elmer) and luciferase activity was measured as relative luminesce units (RLUs) and normalized to values derived from cells infected with pseudotyped virus in the absence of plasma. Data were fit to 2-parameter non-linear regression in Antibody Database[45].

**Reporting summary**. Further information on research design is available in the Nature Research Reporting Summary linked to this article.

## Data availability

The SARS-CoV-2 genomes generated in this study have been deposited in GenBank under accession codes MZ637509-MZ642234 (see Supplementary Data 1) and GISAID (see Supplementary Data 2 for a list of genomes used in phylogenetic analysis and Supplementary Data 3 for genomes used in geographic distribution analysis). The data analyzed as part of this project were obtained from the GISAID database and through a Data Use Agreement between NYC DOHMH and the University of California San Diego. Sequences analyzed by using the vdb tool were downloaded from GISAID. No personally identifying information were included as part of these analyses. Data for Fig. 5 are provided in Supplementary Tables 2 and 3.

## Code availability

The source code for the vdb program is available at the GitHub repository[46]: https://github.com/variant-database/vdb.

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

## Acknowledgements

We thank the Global Initiative on Sharing Avian Influenza Data (GISAID) and the originating and submitting laboratories for sharing the SARS-CoV-2 genome sequences; see Supplementary Data 4 for a list of sequence contributors. We thank Andrew Rambaut and Áine O'Toole for lineage designation. This work was supported by the Caltech Merkin Institute for Translational Research (P.J.B.). This work was supported, in whole or in part, by the Bill & Melinda Gates Foundation [grant INV-002143 to P.J.B.]. Under the grant conditions of the Foundation, a Creative Commons Attribution 4.0 Generic License has already been assigned to the Author Accepted Manuscript version that might arise from this submission. J.O.W. acknowledges funding from the National Institutes of Health (AI135992 and AI136056). T.I.V. is funded by a Branco Weiss Fellowship. M.C.N. is an HHMI Investigator.

## Author contributions

A.P.W., J.O.W., J.L.H., T.I.V., H.H.L., S.H. and J.C.W. analyzed data. J.C.W., M.A.C., E.G. and H.H.L. performed genome sequencing and assembly. J.O.W. curated data. C.G., M. Caskey and M.C.N. provided clinical samples. P.N.P.G. and J.R.K. carried out experiments. A.P.W., C.O.B., Z.Y., S.H., S.S.D., C.E.F., T.I.V, J.L.H. and J.O.W. prepared figures. A.P.W., J.O.W., T.I.V., C.O.B., J.C.W. and S.H. wrote the manuscript with input from all co-authors. A.P.W., P.J.B., J.O.W., J.L.R. and S.H. supervised the study.

## Competing interests

P.J.B. is a co-inventor on a provisional application from the California Institute of Technology for the use of mosaic nanoparticles as coronavirus immunogens. M.C.N., P.J.B. and C.O.B. are co-inventors on provisional applications for several anti-SARS-CoV-2 monoclonal antibodies. J.O.W. has received funding from Gilead Sciences, LLC (completed) and the CDC (ongoing) via grants and contracts to his institution unrelated to this research. All other authors declare no competing interests.

**Additional information**

