## [Peer Review File · Nature Communications]

REVIEWERS' COMMENTS

Reviewer #1 (Remarks to the Author):

My concerns have all been addressed, and I support publication.

Reviewer #2 (Remarks to the Author):

Authors have fulfilled most of the initial requests from the first round of revision.

In response to the reviewers comments the manuscript has improved for clarity and completeness and deserves to be published.

Minor point:

Authors did not test any NTD-specific mAb against B.1.526. Can they comment on whether they expect the NTD mutations are expected to escape?

Response to reviewers:

We have addressed the reviewers' suggestions as follows (reviewers' comments are in gray italics, and our response is in plain black type):

Reviewer #1 (Remarks to the Author):

My concerns have all been addressed, and I support publication.

Reviewer #2 (Remarks to the Author):

Authors have fulfilled most of the initial requests from the first round of revision. In response to the reviewers comments the manuscript has improved for clarity and completeness and deserves to be published.

Minor point:

Authors did not test any NTD-specific mAb against B.1.526. Can they comment on whether they expect the NTD mutations are expected to escape?

Regarding B.1.526 mutations in the N terminal domain (NTD; residues 13-305), mutation D253G has been reported as an escape mutation from antibodies against the N-terminal domain (McCallum, M. *et al. Cell* 184, 2332-2347.e16 (2021)). Hence we would expect some degree of escape from anti-NTD mAbs. This potential escape feature is described in the manuscript at line 93.